# Classical Analog and Hybrid Metamaterials of Tunable Multiple-Band Electromagnetic Induced Transparency

**DOI:** 10.3390/nano12244405

**Published:** 2022-12-09

**Authors:** Zhi Zhang, Duorui Gao, Jinhai Si, Jiacheng Meng

**Affiliations:** 1State Key Laboratory of Transient Optics and Photonics, Xi’an Institute of Optics and Precision Mechanics, Chinese Academy of Sciences, Xi’an 710119, China; 2Key Laboratory for Physical Electronics and Devices of the Ministry of Education and Shaanxi Key Laboratory of Information Photonic Technique, Xi’an Jiaotong University, Xi’an 710049, China; jinhaisi@mail.xjtu.edu.cn; 3School of Optoelectronics, University of Chinese Academy of Sciences, Beijing 100049, China

**Keywords:** harmonic spring oscillation model, metamaterials, electromagnetic-induced transparency

## Abstract

The electromagnetic induced transparency (EIT) effect originates from the destructive interference in an atomic system, which contributes to the transparency window in its response spectrum. The implementation of EIT requires highly demanding laboratory conditions, which greatly limits its acceptance and application. In this paper, an improved harmonic spring oscillation (HSO) model with four oscillators is proposed as a classical analog for the tunable triple-band EIT effect. A more general HSO model including more oscillators is also given, and the analyses of the power absorption in the HSO model conclude a formula, which is more innovative and useful for the study of the multiple-band EIT effect. To further inspect the analogizing ability of the HSO model, a hybrid unit cell containing an electric dipole and toroidal dipoles in the metamaterials is proposed. The highly comparable transmission spectra based on the HSO model and metamaterials indicate the validity of the classical analog in illustrating the formation process of the multiple-band EIT effect in metamaterials. Hence, the HSO model, as a classical analog, is a valid and powerful theoretical tool that can mimic the multiple-band EIT effect in metamaterials.

## 1. Introduction

Recently, the electromagnetic induced transparency (EIT) effect [1,2,3] has been widely studied because of its excellent features. EIT is used to describe a phenomenon where a medium is transformed from opaque to transparent for one probe laser source when it takes in another pump laser source. These laser sources contribute different atomic transitions; hence, they cause the medium to be in the transparent state or to have no energy absorption [4]. Numerous applications based on the EIT effect, such as slow-light devices, optical sensors, and optical switchers, have been reported [5,6,7,8,9,10]. As a quantum effect, EIT is difficult to realize in laboratory conditions and also to understand based on the atomic system. Analog is a useful concept because it allows for analysis and a relation to different phenomena through some common features [11]. Although the effect of quantum physics differs from that of classical physics in both formalism and fundamental notions, the analog can be applied to specific quantum phenomena and their equivalent classical effects [12,13,14]. Two classical analogs of the harmonic spring oscillation (HSO) model and the resistance-inductance-capacitance (RLC) model were created by Alzar to analogize the EIT effect [15], which also provide a theoretical basis for implementing an EIT-like effect in other oscillator systems [16,17]. In the HSO model, Alzar discussed two harmonic oscillators to analogize the probe laser source and the pumping laser source in the atomic system. To make two oscillators in the coupled state, the intrinsic frequencies were set in the near range when their coupling coefficient was nonzero. The analog of the EIT effect was accomplished by observing the transparent windows that emerged in the absorption spectrum of one oscillator. Souza further conducted a more in-depth theoretical study and established a one-to-one correspondence between the HSO model and quantum theory [18]. However, the HSO model is still incomplete and thus unable to analogize the triple-band or multiple-band EIT effect, which would have more application prospects in the multiple-band sensor or switcher. The dynamic coupling process between the oscillators is an issue for demonstrating the tunable EIT state; furthermore, it is unsharp.

Furthermore, metamaterials comprising periodic unit cells are widely used as a plasmon analog of the EIT effect [19,20]. Owing to the excellent electromagnetic characteristics of metamaterials, the EIT effect has several applications in the field of high-sensitivity sensors and energy storage [21,22]. The EIT effect in metamaterials is generally produced by the destructive coupling of a bright and a dark resonant mode [23,24,25]. The dark mode is usually produced by breaking the unit-cell symmetry [26,27]; however, it renders the fabrication process exceptionally complex. Owing to an extremely low loss [28,29,30], a toroidal dipole can also be used to generate the dark resonant mode [31,32]. The toroidal dipole was first demonstrated by Zel’dovich [33], and it is commonly constructed using numerous magnetic dipoles. Hence, the EIT effect in metamaterials can be realized through the conjugation of a low-loss toroidal dipole and a high-loss electric dipole [34,35]. Simultaneously, a tunable EIT effect is generated by changing the oscillation frequency of the electric dipole [36,37,38]. To realize the multiple-band EIT effect, one bright mode should interact synchronously with multiple dark modes [39,40,41]. Furthermore, multiple toroidal dipoles are dependent on the number of different magnetic dipole configurations [42,43]; hence, they are helpful in demonstrating the multiple-band EIT effect [44,45,46]. Although most toroidal dipoles originate from different metamaterial unit cells, both the fabrication process and the modulation are inflexible. The mechanism of the multiple-band bright–dark EIT effect in metamaterials is also not very clear, which should be solved by an analog of the classical physics model.

In this study, to analogize the triple-band EIT effect, we extended and constructed a complex HSO model in order to build three independent interactions among the oscillations. Compared to previous research works [15,18], we classified four oscillators into bright or dark states, depending on their damping rate. The first bright oscillator was connected to the three other dark oscillators with a common junction. This is very similar to the relationship between one probe laser source and three different pumping laser sources in the atomic system. The power absorption spectra of the oscillator displayed three separate transparent windows, hence our model was an appropriate and direct tool for analogizing the multiple-band EIT effect. In addition, a further comprehensive model was proposed by introducing darker oscillators to analogize the multiple-band EIT effect. The most important application of the HSO model is in the development of a formula that describes the power absorption of the oscillators. To the best of our knowledge, the systematic formula was the first theoretical implementation that analyzed the process and the conditions for the occurrence of the multiple-band EIT effect. Simultaneously, to verify the feasibility of the proposed formula in analogizing the multiple-band EIT effect, a kind of hybrid metamaterial was designed by combining a double bar-shaped structure (DD) and a Q-ring-shaped structure (QR) to accomplish a triple-band EIT effect. The DD structure was used to excite an electric dipole, and three toroidal dipoles were stimulated in the QR structure. These structures were identified as the bright and dark modes, respectively, by assessing their transmission spectral bandwidths. Through systematic electromagnetic simulation, the toroidal dipole was primarily explored by comparing the diversity of the magnetic field distribution to determine its origin. Moreover, a tunable triple-band EIT effect was built based on the three toroidal dipoles destructively coupled with a changeable electric dipole. Finally, we synthetically compared the transmission spectra of the HSO model and metamaterials. The spectra were perfectly matched in both coupled and uncoupled states. This demonstrates the feasibility and validity of the HSO model for analogizing the EIT effect in metamaterials.

## 2. Classical Analog Based on Harmonic Spring Oscillations

To construct a triple-band EIT analog, three independent interactions among oscillators should be established [15,18]; hence, we built and extended a new HSO model by introducing four oscillators. The new model is schematically presented in Figure 1a, where four oscillators $m_{j}\left(j=1,a,b,c\right)$ are suspected to be connected to stressed or stretched springs. For the convenience of discussion, the oscillating responses of $m_{j}\left(j=1,a,b,c\right)$ were described as mode j(j=1,a,b,c). One spring with a spring constant $k_{1}$ was attached to a fixed wall on one side of $m_{1}$. The other side of $m_{1}$ was associated with $m_{j}\left(j=a,b,c\right)$ through three independent springs with the spring constant $k_{1j}\left(j=a,b,c\right)$. Equivalently, another three springs, each with a respective spring constant $k_{j}\left(j=a,b,c\right)$, were also connected a common wall. An external harmonic force was acted on $m_{1}$ to make it oscillate actively, and the force behavior was defined as $F_{1s}\left(t\right)=Fe^{-i\omega_{s}t}$. Furthermore, $m_{j}\left(j=a,b,c\right)$ were equally actuated by the transformation springs $k_{1j}\left(j=a,b,c\right)$, leading to synchronous oscillations. The same intrinsic frequency could create two oscillators in the resonant mode, while the oscillating dispersion of one oscillator is controlled by its damping rate. Hence, we made the active oscillator couple with the three other passive oscillators by setting their intrinsic frequencies in the same frequency range. To make the active oscillator couple with each passive oscillator, its damping rate setting had to be much bigger. This provided a wider response from the active oscillator, which could cover other passive oscillators in the frequency domain. The power transferred to the active oscillator is the key factor for the energy transmitted in the HSO model [15], so we characterized a universal formula $x_{j}=N_{j}e^{-i\omega_{s}t}\left(j=1,a,b,c\right)$ to denote the displacement of each oscillator from its balance point. We chose $m_{1}$ as the main target in order to analyze its transferred power by solving the moving equations of all oscillators. Parameter $\eta_{j}\left(j=1,a,b,c\right)$ represented the damping rate of each oscillator $m_{j}$. Thus, the following group of equations were derived based on the discussion on the oscillation of the HSO model: (1)$$m_{1}{\ddot{x}}_{1}=-k_{1}x_{1}-\eta_{1}{\dot{x}}_{1}-k_{1a}\left(x_{1}-x_{a}\right)-k_{1b}\left(x_{1}-x_{b}\right)-k_{1c}\left(x_{1}-x_{c}\right)+F_{1s}\left(t\right)$$
(2)$$m_{a}{\ddot{x}}_{a}=-k_{a}x_{a}-\eta_{a}{\dot{x}}_{a}-k_{1a}\left(x_{a}-x_{1}\right)$$
(3)$$m_{b}{\ddot{x}}_{b}=-k_{b}x_{b}-\eta_{b}{\dot{x}}_{b}-k_{1b}\left(x_{b}-x_{1}\right)$$
(4)$$m_{c}{\ddot{x}}_{c}=-k_{c}x_{c}-\eta_{c}{\dot{x}}_{c}-k_{1c}\left(x_{c}-x_{1}\right)$$

For easy access, the mass of each oscillator was fixed at *m*; therefore, the uniform damping constant was expressed as $\gamma_{j}=\eta_{j}/2m\left(j=1,a,b,c\right)$. The intrinsic frequency of each oscillating mode was set as $\omega_{1}^{2}=\left(k_{1}+k_{1a}+k_{1b}+k_{1c}\right)/m$ and $\omega_{j}^{2}=\left(k_{j}+k_{1j}\right)/m\left(j=a,b,c\right)$, and the coupled oscillating frequency between $m_{1}$ and $m_{j}\left(j=a,b,c\right)$ was set as $\omega_{1j}=k_{1j}/m$. The proposed equations wereare further simplified as follows: (5)$$\left(-{\omega_{s}}^{2}+{\omega_{1}}^{2}-2i\gamma_{1}\omega_{s}\right)N_{1}-\omega_{1a}^{2}N_{a}-\omega_{1b}^{2}N_{b}-\omega_{1c}^{2}N_{c}=\frac{F}{m}$$
(6)$$\left(-{\omega_{s}}^{2}+{\omega_{a}}^{2}-2i\gamma_{a}\omega_{s}\right)N_{a}-\omega_{1a}^{2}N_{1}=0$$
(7)$$\left(-{\omega_{s}}^{2}+{\omega_{b}}^{2}-2i\gamma_{b}\omega_{s}\right)N_{b}-\omega_{1b}^{2}N_{1}=0$$
(8)$$\left(-{\omega_{s}}^{2}+{\omega_{c}}^{2}-2i\gamma_{c}\omega_{s}\right)N_{c}-\omega_{1c}^{2}N_{1}=0$$

We then assumed that the value of $\omega_{s}$ was near to that of $\omega_{j}$, leading to approximations such as ${\omega_{j}}^{2}-{\omega_{s}}^{2}\approx 2\omega_{j}\left(\omega_{j}-\omega_{s}\right)$ and $\gamma_{j}\omega_{s}\approx \gamma_{j}\omega_{j}\left(j=1,a,b,c\right)$ [18]. Similarly, we also introduced the parameter $\Omega_{1s}$ as the driving force coefficient, and $\Omega_{1j}\left(j=a,b,c\right)$ as the coupling coefficient between the oscillator $m_{1}$ and $m_{j}\left(j=a,b,c\right)$ [18]. The parameters were used to describe the coupling state between the active oscillator and each passive oscillator in the HSO model, which can be used to control the coupling strength.
(9)$$\Omega_{1s}=\sqrt{\frac{F^{2}}{2m\omega_{1}}},\Omega_{1a}=\frac{\omega_{1a}^{2}}{2\sqrt{\omega_{1}\omega_{a}}},\Omega_{1b}=\frac{\omega_{1b}^{2}}{2\sqrt{\omega_{1}\omega_{b}}},\Omega_{1c}=\frac{\omega_{1c}^{2}}{2\sqrt{\omega_{1}\omega_{c}}}$$

*K*,*M*, and $C_{1}$ were proposed to deformalize the equations.
(10)$$K=\left(\omega_{1}-\omega_{s}-i\gamma_{1}\right)\left(\omega_{a}-\omega_{s}-i\gamma_{a}\right)\left(\omega_{b}-\omega_{s}-i\gamma_{b}\right)\left(\omega_{c}-\omega_{s}-i\gamma_{c}\right)$$
(11)$$\begin{array}{c} M={\Omega_{1a}}^{2}\left(\omega_{b}-\omega_{s}-i\gamma_{b}\right)\left(\omega_{c}-\omega_{s}-i\gamma_{c}\right)-{\Omega_{1b}}^{2}\left(\omega_{a}-\omega_{s}-i\gamma_{a}\right)\left(\omega_{c}-\omega_{s}-i\gamma_{c}\right)\\ -{\Omega_{1c}}^{2}\left(\omega_{a}-\omega_{s}-i\gamma_{a}\right)\left(\omega_{b}-\omega_{s}-i\gamma_{b}\right) \end{array}$$
(12)$$C_{1}=\sqrt{\frac{1}{2m\omega_{1}}}$$

Then, we deduced $x_{1}$ as follows: (13)$$x_{1}\left(t\right)=\frac{C_{1}\Omega_{1s}\left(\omega_{a}-\omega_{s}-i\gamma_{a}\right)\left(\omega_{b}-\omega_{s}-i\gamma_{b}\right)\left(\omega_{c}-\omega_{s}-i\gamma_{c}\right)}{K-M}e^{-i\omega_{s}t}$$

Moreover, the equation above was be rewritten as follows: (14)$$x_{1}\left(t\right)=C_{1}\rho_{s}e^{-i\omega_{s}t}$$
(15)$$\rho_{s}=\frac{\Omega_{1s}\left(\omega_{a}-\omega_{s}-i\gamma_{a}\right)\left(\omega_{b}-\omega_{s}-i\gamma_{b}\right)\left(\omega_{c}-\omega_{s}-i\gamma_{c}\right)}{K-M}$$

The devised $\rho_{s}$ refers to the concept of mechanical susceptibility [18], which is a time-independent part of $x_{1}\left(t\right)$. Its real part can be used to describe the energy absorption of an oscillator, while its imaginary part is related to energy dispersion. It is crucial for assessing the oscillating state of $m_{1}$, and the oscillation response of $m_{j}\left(j=1,a,b,c\right)$ is related to the frequency deviation of $\omega_{j}-\omega_{s}\left(j=1,a,b,c\right)$ and the oscillation damping rate $\gamma_{j}\left(j=1,a,b,c\right)$ in Figure 1b. These oscillating parameters were quite important in making the HSO analogize the EIT effect because their values determine the behaviors of each oscillator. The spectra of $1-Im(\rho_{s})$ illustrates the power transmission of $m_{1}$ when the external harmonic frequency is varied. When the intrinsic frequency of $m_{1}$ was not located in the same frequency range, the whole system was in the uncoupled state. Conversely, in the coupled state, the initial frequency $\omega_{s}\left(j=a,b,c\right)$ was set at 1.5, 2.0, and 2.5 THz, respectively. $\omega_{1}$ and $\gamma_{1}$ were set at 2.0 THz and 0.05, which enabled the oscillating response of $m_{1}$ to cover $m_{j}\left(j=a,b,c\right)$ in the frequency domain. The spectra showed three separate transparent windows at the intrinsic frequency of $m_{j}\left(j=a,b,c\right)$ when the HSO model was in the coupled state, which was just the analog of the triple-band EIT effect. Furthermore, to investigate how the damping rate could determine the oscillating response, $\gamma_{1}$ was set at 0.05 and $\gamma_{j}\left(j=a,b,c\right)$ was set at 0.0001. Accordingly, the dip bandwidth of mode 1 in Figure 1b was much wider than that of other modes. Subsequently, the value of $\gamma_{j}\left(j=1,a,b,c\right)$ was additionally reset to 0.001 and 0.01, respectively, and their spectral amplitude was subtracted from 0.1 and 0.2, respectively, to distinguish them from each other. The spectra transformations showed that the bandwidth of mode a, b, and c became wider with $\gamma_{j}$ increasing from 0.0001 to 0.01. Hence, the damping rate can be used to define the mode as a bright mode or a dark mode.

To render the application of the HSO model universal, we deduced and assumed the generalized formation Equation (Equation 16) by imitating Equation (Equation 15). The more general HSO model of one active oscillator interacting with the $n_{th}$ passive oscillator was used to demonstrate the multiple-band EIT effect. In Equation (Equation 16), $\omega_{0}$ is the intrinsic frequency of the active oscillator, and $\omega_{i}\left(i=1,2,3,...,n\right)$ is the intrinsic frequency of each passive oscillator. The number of dark modes introduced into the HSO model was dependent on the number of passive oscillators. To examine the validity of Equation (Equation 16), we set *n* as four and five, and the spectra displayed four or five narrow dips when the modes were in the uncoupled state. In the coupled state, there were four transparent windows in the spectra when *n* was set as four. Consequently, Equation (Equation 16), which is supported by the HSO model, is a universal and useful tool for validating the multiple-band EIT effect.
(16)$$\rho_{ns}=\Omega_{1s}\frac{\prod_{i=1}^{n}\left(\omega_{i}-\omega_{s}-i\gamma_{i}\right)}{\left(\omega_{0}-\omega_{s}-i\gamma_{0}\right)\prod_{i=1}^{n}\left(\omega_{i}-\omega_{s}-i\gamma_{i}\right)-\sum_{i=1}^{n}\Omega_{1i}^{2}\prod_{j\neq i}^{n-1}\left(\omega_{j}-\omega_{s}-i\gamma_{j}\right)}$$

Moreover, we checked how the the coupling coefficients $\Omega_{1}$ and $\Omega_{1j}\left(j=a,b,c\right)$ impact the spectrum-changing tendency. In Figure 1c, $\Omega_{1j}\left(j=1,a,b,c\right)$ increases from 0.005 to 0.12, whereas $\Omega_{1}$ remains constant at 0.05. The transmission attenuation values of modes a, b, and c were entirely in reductive development, with some fluctuations. This illustrates that a strong coupling rate can lead to a high resonant strength in the oscillator. More specifically, this parameter can be used to balance the resonant intensities of multiple dark modes. However, the resonant frequency also shifted to a high range when the coupling rates continuously increased, as shown in Figure 1c. Overly strong coupling coefficients caused the resonant frequencies of the passive oscillators to deviate from their intrinsic frequencies. Thus, it is essential to select appropriate values to define the HSO model. The above discussion is meaningful for revealing the applicability of Equations (15) and (16) in determining the resonant behavior in the HSO model.

## 3. Structure Design and Simulation

Metamaterials are a good way of demonstrating the properties of the EIT effect; therefore, we proposed a metamaterial structure that could accomplish the triple-band EIT effect. A schematic of the metamaterial structure is illustrated in Figure 2a, where there are two layers of unit cells made of lossy gold (Au; yellow area) with an electrical conduc-tivity of 4.561 × 10${}^{7}$ S/m. For the convenience of discussion, the lower unit cell is named the QR structure owing to its Q-ring configuration, and the upper unit cell is named the DD structure owing to its double-bar shape. They were combined to form a hybrid structure called the “QR-DD”, which was located on a quartz substrate (blue area) with a dielectric permittivity of 3.75. The separated medium made of polytetrafluoroethylene had a dielectric permittivity of 2.1. The geometric parameters of the XY plane in Figure 2a were L1 = 50 μm, W1 = 60 μm, P11 = P12 = 50 μm, W2 = 60 μm, P21 = P22 = 50 μm. L2 varied between 30 μm and 60 μm, and the thicknesses of each layer on the XZ plane were t1 = t2 = 0.2 μm, SP = 10 μm, and SUB = 50 μm. We utilized the finite-difference time-domain method to explore the resonant behaviors in the QR-DD structure. The Y-axis polarization wave normally excited the proposed unit cell in the terahertz regime. The periodic boundary condition was set in both the X and Y-axes, with an open boundary condition along the Z-axis. The bright mode was stimulated by the electric dipole in the DD structure, which was primarily dependent on the bar length. The dark mode was excited by the toroidal dipolar resonance in the QR structure; however, its origin required further investigation. Hence, we studied the transmission spectra based on the QR structure in Figure 2b. Three narrow dips were visible, indicating that there were three dark modes in the QR structure. For convenience, they are denoted as T1, T2, and T3, respectively. To directly investigate the mechanism of these dark modes, the patterns of the Y-axis electric field and the Z-axis magnetic field are simultaneously displayed in Figure 2c when the QR structure was stimulated at 2.403, 2.471, and 2.562 THz. These frequencies correspond to the frequencies of the dark modes shown in Figure 2b. Correspondingly, the Z-axis magnetic fields in the structure were generated by the currents flowing in the Y-axis loop according to the Biot–Savart law. In each pattern, we assumed that an asymmetric magnetic distribution could create a spinning field circulating along the Y-axis loop on the YZ plane. The vortical magnetic field contributed to the magnetic dipole. Hence, according to the right-hand rule, a magnetic toroidal dipole composed of several magnetic dipoles was located along the Y-axis loop. Naturally, the three different magnetic distribution patterns excited by different concentrated areas of currents are analogous to the three toroidal dipoles in Figure 1c.

To confirm the above conjecture, five structural transformations, A1–A5, from the QR structure are shown in Figure 3c. The A1 and A3 structures were reduced by 4 μm in width from the inward and outward sides of the Y-axis loop, respectively. The A2 structure was reduced by 2 μm in width from both sides, whereas the A4 structure was reduced by 4 μm in width from the inward side of the X-axis. As reference, the A5 structure remained unchanged. The resonant frequency shifts of the three dark modes in Figure 3a were not particularly noticeable, indicating that the toroidal dipoles are primarily dependent on the overall geometric dimensions. However, if we compare the resonant strengths of the five structures in Figure 3b, the resonant strength of the toroidal dipoles can be considered to be abandoned or extremely weak. Based on the magnetic and electric field distributions in Figure 2c, the first-order toroidal dipole (T1) was mainly induced by vortical magnetic fields circling along the outer edge of the Y-axis loop. In Figure 3b, the low transmittance of the T1 mode was extremely weak because a small part of the A3 structure was just on the outward edge of the Y-axis loop. By using the same method to analyze the A1 structure, the T2 mode was shown to almost exist owing to the inward-lacking edge of the Y-axis loop. This illustrates that the toroidal dipolar mode is strongly dependent on the magnetic field circling area. Furthermore, the T1 and T2 modes in the A2 structure still existed but were weaker than those in the A5 structure. Hence, we assumed that the A2 structure contributes to the formation of the A5 structure, which can also achieve the T1 and T2 modes at both edges of a new Y-axis loop. However, the resonant strength decreased when the Y-axis loop width decreased. The absent part of the A4 structure was the inward edge of the X-axis loop, and there was no obvious distinction compared with the A5 structure in Figure 3b. This indicates that the X-axis magnetic field distribution had no influence on the toroidal dipolar resonance. The toroidal dipole depends on the magnetic field distribution in the Y-axis direction, which is parallel to the polarization of the incident wave. The discussions above represent the origin of the three toroidal dipoles in the QR structure; therefore, these dipoles can individually act as dark modes by designing the structural profile.

To further elucidate the mechanism of the triple-band EIT effect, we investigated the coupling process between the bright and dark modes in the QR-DD structure. As previously discussed, the electric dipole excited the bright mode in the DD structure, whereas the dark modes were produced by the toroidal dipoles in the QR structure. To ensure that the dark modes would remain unchanged, the overall geometric dimensions of the QR structure were fixed. The L2 was set at 60 μm or 45 μm to vary the bright mode in the DD structure from 2.1 THz to 2.4 THz. The four transmission spectra based on the four separate structures are shown in Figure 4a. They were taken from the QR structure; from the hybrid QR-DD structure when L2 was set at 45 μm; from the hybrid QR-DD structure when L2 was set at 60 μm; and from the DD structure when L2 was set at 60 μm. Hence, for simplicity, we marked each spectrum as QR, QR-DD 45, QR-DD 60, and DD 60, respectively. The broad first-order dips in QR-DD 60 and DD 60 were the responses of the bright mode in the DD structure. Additionally, three narrow dips in QR and QR-DD 60 appeared at approximately 2.37, 2.43, and 2.52 THz, which matched the three dark adjacent modes in the QR structure. Concurrently, the dark modes shifted by approximately 30–42 GHz in the spectral comparison between QR and QR-DD 60; similarly, the bright mode shifted by approximately 78 GHz between QR-DD 60 and DD 60. These resonant shifts illustrate that the couplings of the bright and dark resonant modes affect each other. When L2 was set at 60 μm in the QR-DD structure, the bright mode could not cover the dark modes. On the contrary, when we set L2 at 45 μm to make the bright mode resonate at approximately 2.4 THz, a triple-band EIT effect could be observed in the QR-DD 45 structure. The creation of the three transparent windows indicates that the coupling state between the bright and dark modes is the key factor in producing the triple-band EIT effect. The current flow distribution in Figure 4b1,b2 shows the power absorption deviations in the QR-DD structure. When L2 was set at 60 μm, no EIT effect was observed, as shown in Figure 4b1; hence, the excited currents converged around the structure. By contrast, when L2 was set at 45 μm, the triple-band EIT effect rendered the entire structure transparent at 2.52 THz, and only a few currents existed. Additional observations of the Z-axis excited by the magnetic field distribution in the two structures provide another interpretation. In Figure 4b3,b5, the frail magnetic field indicates that there is no bright mode in the DD structure at 2.52 THz, whereas the dark mode can be excited in the QR structure. The undesirable magnetic fields lead to sharp current congestion, as shown in Figure 4b1. For comparison, the bright and dark modes were both stimulated at 2.52 THz in Figure 4b4,b6. Consequently, the lower absorption in Figure 4b2 is a result of the destructive interaction between the bright and dark modes, which directly produced the EIT effect. A similar analytical process is also appropriate for when the structure was excited at 2.37 THz and 2.43 THz. From the above discussion, we can deduce that a triple-band EIT effect is produced when the bright mode has the ability to cover all dark modes in the frequency domain.

To verify the mimicking ability of the HSO model, we summarized the spectral data based on the QR-DD and HSO models in Figure 5. There are several common features in the resonant responses when comparing the spectral profiles, and the imperceptible numerical difference between them is sufficient to prove the validity and fitness of the HSO model.

In the HSO model, the intrinsic frequencies $\omega_{j}\left(j=a,b,c\right)$ of the passive oscillator were reset to 2.3, 2.43, and 2.57 THz, respectively, and these frequencies corresponded to the dark modes in the QR-DD structure. Subsequently, when $\omega_{0}$ was set at 2.5 THz, three transparent windows were medially present at 2.37, 2.43, and 2.53 THz, respectively. This phenomenon was because of the bright mode overlapping the dark modes, which caused the HSO model to be in the EIT state. When $\omega_{0}$ shifted to 2.1 THz, the transparent windows disappeared again, and the whole HSO model was converted into an uncoupled status. Such a conversion action is highly analogous to the transmission spectra of the QR-DD structure because L2 can control the bright mode to couple with three dark modes. For the coupled situation, three transparent windows in the metamaterials were also centrally present at approximately 2.37, 2.43, and 2.53 THz. In the uncoupled situation, only three low-loss resonant dips were observed in the range of 2.3–2.6 THz. Overall, the HSO model is a highly useful numerical tool for mimicking the EIT effect in metamaterials. Predictably, the HSO model is also useful in deriving the analog of the multiple-band EIT effect by using Equation (Equation 16).

## 4. Conclusions

In summary, we introduced an extended HSO model with four oscillators; one externally driven active oscillator can interact with three other passive oscillators. A series of functions characterizing the resonant response was mathematically deduced based on the HSO model, and the parameters in the equations required to demonstrate the impact of the resonance were also identified. For example, the dark and bright modes depended heavily on the damping rate, and a stronger coupling coefficient could shift the oscillation frequency and enhance resonant strength. Based on the energy transmission spectra, the HSO model was applied to analogize the triple-band EIT effect when the bright mode was coupled with three dark modes. The number of dark modes is the key factor in generating the multiple-band EIT effect; therefore, a more comprehensive HSO model involving a higher number of dark modes is also a suitable analog of the multiple-band EIT effect. To further demonstrate the applicability of the HSO model, a carefully designed hybrid metamaterial structure was used to show the triple-band EIT effect at 2.37, 2.43, and 2.52 THz. This is because of the destructive coupling of the electric dipole and the three toroidal dipoles. The mechanism of the toroidal dipole was discussed in detail by exploring the magnetic field and current flow distribution in a transformable QR structure. We discovered and proved that three coterminous toroidal dipoles were produced in diverse regions of the QR structure. We defined the mode as bright or dark, depending on the bandwidth of the resonant response. According to the spectra of the systematic simulation, the coupled state is dependent on whether the bright mode is stimulated in the same frequency range as the dark mode, which supports the possibility of generating a triple-band EIT effect. Finally, the comparison of the transmission spectra of the QR-DD structure and the HSO model proved that the proposed classical analog is well-suited to analogize the formation of the triple-band EIT effect in metamaterials. This classical model can effectively guide the construction and implementation of a tunable multiple-band EIT effect in metamaterials.

## Figures and Tables

**Figure 1 nanomaterials-12-04405-f001:**
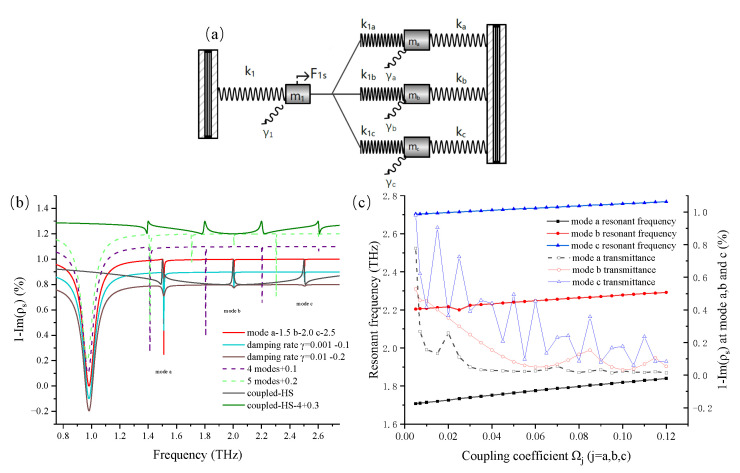
(**a**) A schematic of the modified HSO model with four oscillators. The model is composed of four oscillators, namely $m_{1}$, $m_{a}$, $m_{b}$, and $m_{c}$. They are connected to each other and to walls using seven springs with different spring constants, $k_{j}\left(j=1,a,b,c,1a,1b,1c\right)$. A driven force $F_{1s}\left(t\right)=Fe^{-i\omega_{s}t}$ acts on $m_{1}$, and $m_{1}$ oscillates with a damping rate of $\gamma_{1}$ under the combined action of $F_{1s}$ and other connected springs. Correspondingly, the other three oscillators also oscillate passively owing to their connected springs with a damping rate of $\gamma_{j}\left(j=a,b,c\right)$. (**b**) Illustration of the spectra of $1-Im(\rho_{s})$ with varying parameters. The initial frequency $\omega_{j}\left(j=1,a,b,c\right)$ identifies the position of the transmission dip in the spectra, and the damping rate influences the resonant response. The EIT effect is related to whether the initial frequency and the response bandwidth of each mode are overlapped. The influence of the damping rate on the resonant response is also compared, and the amplitude of the spectrum is subtracted from 0.1 and 0.2 for a clear distinction. A comparison of the spectra based on the HSO system with four or five passive modes in the uncoupled state is shown. Their amplitudes are added with 0.1 and 0.2 to distinguish them from other spectra. A tetradic band EIT effect in the spectra when the active mode couples with four other passive modes in the HSO model. (**c**) The changing trend of $1-Im(\rho_{s})$ decreases and the resonant frequency gradually increases when the coupling coefficient $\Omega_{j}\left(j=a,b,c\right)$ increases from 0.005 to 0.12.

**Figure 2 nanomaterials-12-04405-f002:**
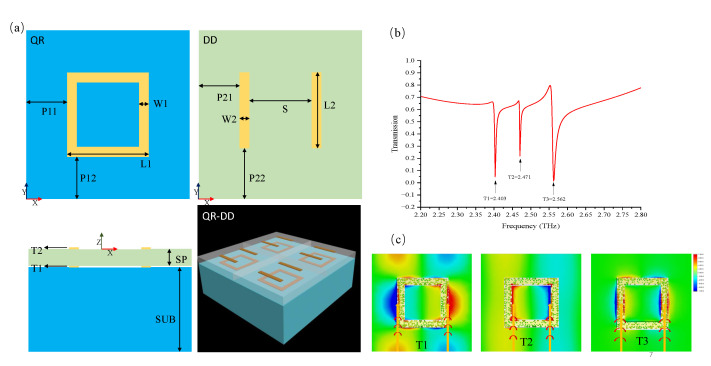
(**a**) Schematics of the QR and DD structures are shown, and the dimension parameters are as follows: L1 = 50 μm, W1 = 60 μm, P11 = P12 = 50 μm, W2 = 60 μm, P21 = P22 = 50 μm, t1 = t2 = 0.2 μm, SP = 10 μm, and SUB = 50 μm. L2 is tunable in the range of 30–60 μm. (**b**) The electromagnetic simulation of the transmission spectra for the QR structure shows three distinct narrow dips at 2.403, 2.471, and 2.562 THz. For convenience, these resonant modes are marked as dark mode T1, T2, and T3, respectively. (**c**) The Y-axis electric-field distribution and the Z-axis magnetic field patterns of different modes are shown. The concentrated area of currents generates disparate magnetic fields; therefore, an asymmetric magnetic distribution produces three diverse vortical magnetic fields. The three magnetic dipoles are stimulated along the Y-axis loop in each pattern. They contribute to the three magnetic toroidal dipoles indicated by three vertical arrows.

**Figure 3 nanomaterials-12-04405-f003:**
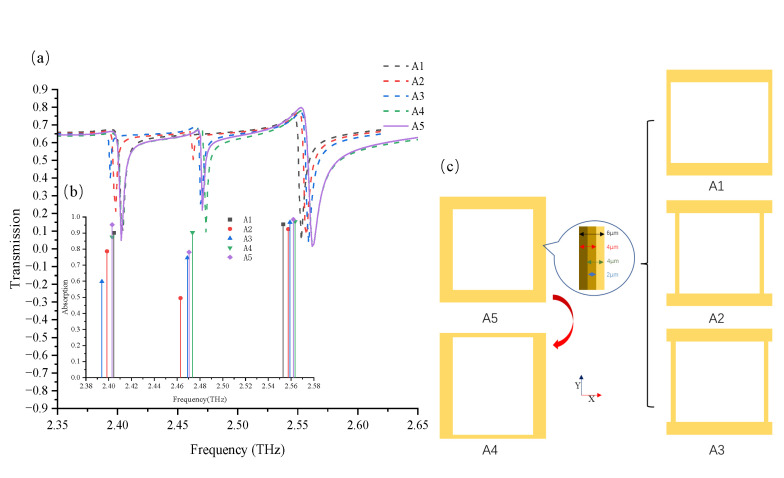
(**a**) Comparison of the simulation transmission spectra of the A1, A2, A3, A4, and A5 structures. (**b**) Comparison of the absolute values of transmission attenuation when the A1, A2, A3, A4, and A5 structures are stimulated in the T1, T2, and T3 modes, respectively. (**c**) Schematics of the A1, A2, A3, A4, and A5 structures transformed in the QR structure. These structures were reduced by 4 μm in width on the inward side of the Y-axis loop (A1), 2 μm in width on both sides of the Y-axis loop (A2), 4 μm in width on the outward side of the Y-axis loop (A3), 4 μm in width on the inward side of the X-axis loop (A4), and remained unchanged, which was used as a reference (A5).

**Figure 4 nanomaterials-12-04405-f004:**
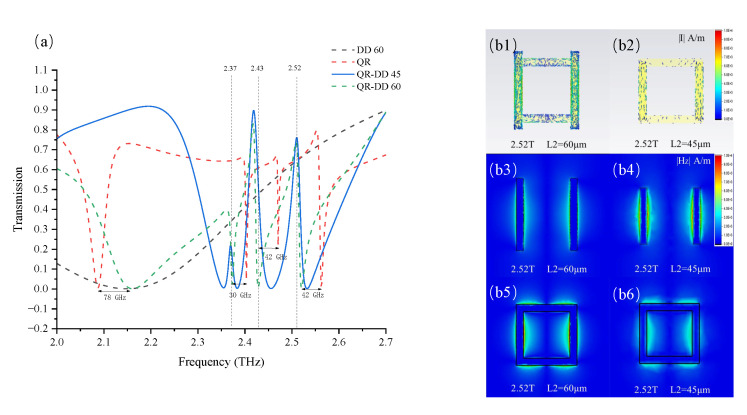
(**a**) A comparison of the simulation transmission spectra based on four different structure combinations is shown; they are marked as DD 60, QR, QR-DD 45, and QR-DD 60. (**b1**,**b2**) The conversions of the Y-axis current flow and the density distribution when the QR-DD structure is excited at 2.52 THz and when L2 is set at 60 μm or 45 μm, respectively. The current convergence situation varies depending on whether the variable L2 transforms the QR-DD structure into an EIT state. (**b3**–**b6**) The conversions of absolute Z-axis magnetic field distributions when the two structures are excited at 2.53 THz and the L2 in the structure is set at 60 μm or 45 μm. The difference in the magnetic field distribution explains the variability of the excitation mode, and the interaction between the two modes leads to the EIT phenomenon.

**Figure 5 nanomaterials-12-04405-f005:**
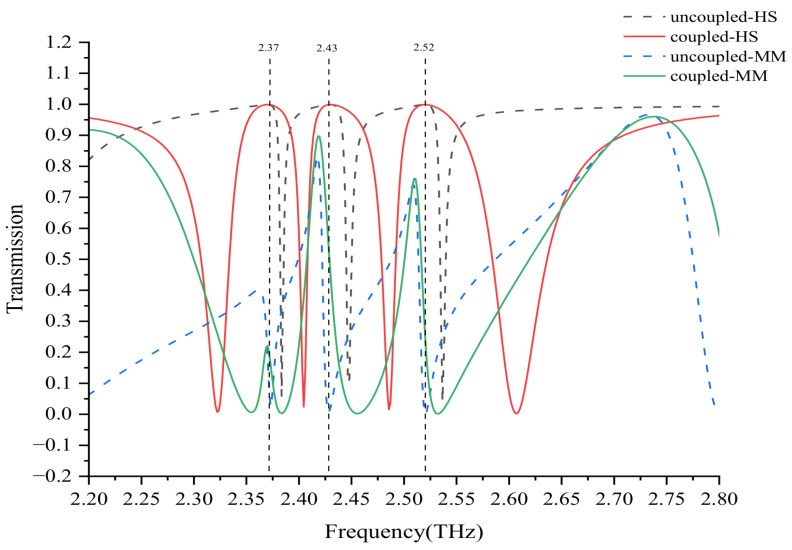
Comparisons of the comprehensive spectra based on the QR-DD structure and HSO model when the bright and dark modes are in the coupled and uncoupled states, respectively. Transparent windows are formed in the spectra when the bright and dark modes of the two models interact in a destructive manner. If the modes are in an uncoupled state, the bright and dark modes exist separately in the spectra.

## Data Availability

Data sharing is not applicable to this article.

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
