# Peer review of "Classical Analog and Hybrid Metamaterials of Tunable Multiple-Band Electromagnetic Induced Transparency"

_nanomaterials, 2022, doi:10.3390/nano12244405_

Round 1

Reviewer 1 Report

Based on the classical harmonic oscillator model  for the  electromagnetic induced transparency  developed and reported previously by other authors,  the authors of the reviewed paper extended it to the case of four oscillators. The most interesting result is that they suggested and worked out a new device design that, in turn, according to performed theoretical calculations, is expected to be able to possess desirable properties and can be proved experimentally.

Questions and comments:

1.      Line 136-137- It remained unclear the argumentation of the choice of coupling coefficient values.

2.      Line 168, 173.  Figure 1b seems to be typos (instead of Figure 2b)

3.      Figure 1 has to be placed closer to its explanation in the text. Figure size and text fonts are too small. 

F

F

Reviewer 2 Report

In their paper "Classical analog and hybrid metamaterials of tunable multiple-band electromagnetic induced transparency" the authors consider the effect of electromagnetically induced transparency (EIT) arising from destructive interference in an atomic system. For the study the authors chose the model of coupled four harmonic oscillators. As stated by the authors, that the well-comparable transmission spectra based on the HSO model indicate the validity of this classical analogue for illustrating the formation process of the multiband EIT effect in metamaterials. Consequently, the HSO model, as a classical analogue, is correct.

The model under study is quite simple and straightforward. The only question I have, and as it seems to me the authors should seriously answer it. Why this model of four connected oscillators is physically realizable?  It is hard to imagine how a real system must behave in order to fit this model. Mathematically you can choose any model and explain probably anything by these models, but how real they will be is a question worthy of quite clear argumentation. The authors should more clearly and in detail argue the choice of this model to explain the effect of electromagnetically induced transparency.

Round 2

Reviewer 2 Report

Thank you for such a detailed explanation, sometimes too detailed. My question was simple enough. What real systems can be four coupled oscillators and why? I didn't see this answer. While your answer is quite detailed, it is not entirely complete. For a better understanding by the reader of the chosen model, it is necessary to briefly (from 5 to 10 sentences) explain your choice.
